# WHEN TREATMENT EFFECT ESTIMATION MEETS COLLIDER BIAS: A DUAL COUNTERFACTUAL GENERATIVE APPROACH

## ABSTRACT

Collider bias poses a great challenge in estimating the treatment effect from observational data due to the sample selection mechanism on both treatments and outcomes. Previous works mainly focused on addressing confounding bias and selection bias caused by covariates only. However, they failed to accurately estimate the causal effect with collider bias, which is known to be an unidentifiable problem without further assumptions on the observational data. In this paper, we address collider bias in the observational data by introducing small-scale experimental data. Specifically, we treat the collider bias problem from an out-of-distribution perspective, where the selected observational data comes from an environment labeled with $S = 1$, and the unselected data comes from another environment labeled with $S = 0$. The experimental data comes from the entire data space, but the environment labels are unknown. Then, we propose a novel method named Dual Counterfactual Generative Model (DCGM), which consists of two generators that respectively generate the unselected data and the missing $S$ labels, and two discriminators that discriminate between the observational data and data with generated $S = 1$ labels, as well as between the generated unselected samples and data with generated $S = 0$ labels for training the generators. Combining the observational data with the unselected samples generated by DCGM, the treatment effect can be accurately estimated using the existing approaches without considering the collider bias. Extensive experiments on synthetic and real-world data demonstrate the effectiveness and the potential application value of the proposed method.

## 1 INTRODUCTION

Estimating treatment effects from observational data is crucial for explanatory analysis and decision-making processes (Robins et al., 2000; Angrist & Pischke, 2009; Imbens & Wooldridge, 2009; Emdin et al., 2017). For example, accurately assessing the treatment effect of specific drugs on each patient can help doctors decide how to administer drugs to specific individuals, which is a counterfactual problem since we cannot simultaneously observe the outcomes of an individual taking or not taking the drugs. The critical challenge of estimating treatment effects is eliminating the presence of biases in the observational data (Pearl, 2009).

There are two primary sources for biases: confounding bias and selection bias (Greenland, 2003; Guo et al., 2020; Hernán & Robins, 2020). Let $T$ denote the treatment variable, $\mathbf{X}$ denote the pre-treatment variables, $Y$ denote the outcome variable, and $S$ denote the selection indicator. The confounding bias results from common causes of treatments and outcomes ($T \leftarrow \mathbf{X} \rightarrow Y$), and the selection bias results from non-random sample selection caused by some certain variables ($T \dashrightarrow S \dashleftarrow Y$). Most of the previous works focused on addressing confounding bias (Bang & Robins, 2005; Shalit et al., 2017; Louizos et al., 2017; Wager & Athey, 2018) and selection bias caused by only $T$ and $\mathbf{X}$ (Bareinboim & Tian, 2015; Correa et al., 2018), while ignoring collider bias which is a particular form of selection bias ($T \rightarrow S \leftarrow Y$). These methods cannot address collider bias because both $T$ and $Y$ cause $S$, which introduces spurious correlations between $T$ and $Y$, resulting in biased estimation of treatment effects not only on $S = 0$ data but also on $S = 1$ data.

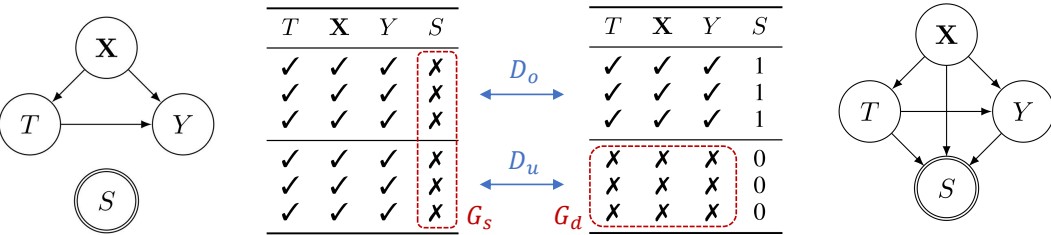

(a) Experimental data without collider bias   (b) Observational data with collider bias

Figure 1: The data form and causal graphs of observation data and experimental data where ✓ denotes the data is observable and ✗ denotes the data cannot be observed.

Collider bias can be defined as non-random sample selection conditioning on both treatments and outcomes, as shown in Figure 1(b). The observational data is sampled from the true data distribution by the sample selection mechanism in Figure 1(b), indicated as $S = 1$, while the unobserved non-selected data is indicated as $S = 0$. In other words, only $S = 1$ samples can be observed, and for $S = 0$ data, the values of $\mathbf{X}$, $T$ and $Y$ are all missing. Due to collider bias, the observed data distribution will differ from the true data distribution. For example, when studying whether vaccination will protect against contracting COVID-19, where $T$ is whether an individual is vaccinated, $Y$ is whether an individual contracts COVID-19, and $\mathbf{X}$ is an individual's covariates like gender, age, etc., we cannot force everyone to test for COVID-19. As a result, we can only observe the data of a specific population who test for COVID-19. However, whether testing for COVID-19 is not random, people who are vaccinated and who contract COVID-19 are more willing to test, which means the sample selection is conditional on the values of $T$ and $Y$, leading to collider bias. In fact, without further assumptions about the observational data, treatment effects are unidentifiable with collider bias (Correa & Bareinboim, 2017; Hernán & Robins, 2020), and thus it is necessary to introduce external unbiased data to solve collider bias.

Fortunately, we can conduct small experimental studies on randomly selected units in real-world applications. For instance, online ticketing platforms can provide incentives or rewards for randomly selected users to rate the movies. This helps to mitigate only the collider bias in the experimental data, which means that the observed covariates in this dataset can be regarded as the representative units of the entire population, and the collected experimental dataset has the same data distribution in the sense that $\mathbb{P}(\mathbf{X})$ remains the same. However, conducting such random experiments is expensive. As a result, when estimating heterogeneous treatment effects, using only the experimental data is insufficient because of the severe overfitting problem. Nevertheless, combining a small-scale experimental dataset with a large-scale biased observational dataset to address collider bias is feasible.

In this paper, we present a novel formulation of collider bias as an out-of-distribution problem, as illustrated in Figure 1. Specifically, we treat the selection indicator $S$ as the environment label, such that the observational data and the unselected data, respectively, come from an environment labeled with $S = 1$ and $S = 0$, and the experimental data is derived from the entire data space, but with unknown environment labels. Therefore, to address this challenge, we propose using both datasets

- to generate the missing $S = 0$ samples of the observational dataset,
- to generate the missing $S$ labels in the experimental dataset,
- to align the distribution of the combined generated $S = 0$ samples with the observational dataset to match that of the entire data space.

To achieve the above objectives, we propose a novel method named **D**ual **C**ounterfactual **G**enerative **M**odel, called **DCGM**, which consists of two generators that respectively generate the missing $S = 0$ samples and the missing $S$ labels, as well as two discriminators that distinguish between the observational data and data with generated $S = 1$ labels, and between the generated unselected samples and data with generated $S = 0$ labels. By optimizing the generators using the discriminators, DCGM can effectively generate missing data while preserving the original data distribution. Combining the observational data with the unselected samples generated by DCGM, we can flexibly use any existing treatment effect estimation methods to achieve an accurate estimate. Extensive experiments on synthetic and real-world datasets have demonstrated the effectiveness of DCGM. By plugging DCGM

into various treatment effect estimators, we have achieved significant improvements, outperforming existing state-of-the-art methods.

## 2 RELATED WORKS

Previous works on confounding bias in observational studies include propensity-score-based, confounder balancing, tree-based, representation-learning-based, and generative-model-based methods. The propensity score was introduced in (Rosenbaum & Rubin, 1983) and defined as $\mathbb{P}(T = 1 \mid \mathbf{X} = \mathbf{x})$. Based on the propensity score, various estimators have been proposed, such as propensity score matching (Dehejia & Wahba, 2002), Inverse Probability of Treatment Weighting (IPTW) (Hirano et al., 2003), and the doubly robust estimator that combines IPTW with regression (Bang & Robins, 2005). Confounder balancing is to learn sample weights that make the confounder distributions of control and treated units similar through sample re-weighting, such as Entropy Balancing (Hainmueller, 2012) and Approximate Residual Balancing (Athey et al., 2018). Tree-based methods like Causal Forest (Wager & Athey, 2018) build a large number of causal trees with different sub-sampling rates and then estimate heterogeneous treatment effects by taking an average of the outcomes from these causal trees. Methods based on deep representation learning were proposed to learn a balanced representation of covariates, such as Treatment-Agnostic Representation Network, Balancing Neural Network (Johansson et al., 2016), Counterfactual Regression (Shalit et al., 2017) and Disentangled Representations for CounterFactual Regression (Hassanpour & Greiner, 2020). Generative methods include CEVAE (Louizos et al., 2017) that applies variational autoencoders to address hidden confounders, and GANITE (Yoon et al., 2018) generates counterfactual outcomes and ITEs. Detailed discussion on the difference between our proposed method and previous generative-model-based methods is in Section A.3.

If we are only interested in the treatment effect of $S = 1$ data, all the above methods can also deal with selection bias that only leads to spurious correlations between $\mathbf{X}$ and $T$, which has a similar impact on treatment effect estimation to confounding bias. However, they cannot deal with the more general scenario that we also need to estimate the treatment effect of $S = 0$ data. Previous works on selection bias mainly focus on sample selection caused by only $\mathbf{X}$ and $T$. Suppose there are variables in the causal graph that satisfy the selection-backdoor criterion. In that case, selection bias can be addressed by selection-backdoor adjustment (Bareinboim et al., 2014; Bareinboim & Tian, 2015; Correa & Bareinboim, 2017; Correa et al., 2018). However, these methods cannot solve collider bias because no valid adjustment can block the non-causal path $T \to S \leftarrow Y$, which directly introduces spurious correlations between $\mathbf{X}$ and $Y$. In fact, without further assumptions about the observational data, treatment effects are unidentifiable with collider bias (Correa & Bareinboim, 2017; Hernán & Robins, 2020). To the best of our knowledge, there are currently no methods to solve collider bias without making further assumptions about the observational data.

## 3 PROBLEM AND ALGORITHM

### 3.1 PROBLEM FORMULATION

Let $\mathcal{D} = \{\mathbf{x}_i, t_i, y_i\}_{i=1}^n$ be a sample population with $n$ units independently drawn from the true target data distribution $\mathbb{P}$. For a unit $i$, $t_i \in \{0, 1\}$ is the binary treatment, $y_i$ is the outcome, and $\mathbf{x}_i \in \mathbb{R}^{d \times 1}$ is the observed pre-treatment variables with $d$ dimensions. We have a large-scale dataset of observational samples non-randomly drawn from $\mathbb{P}$, denoted as $\mathcal{D}_{\text{obs}}$, and a small-scale experimental dataset $\mathcal{D}_{\text{exp}}$ conducted on units randomly sampled from $\mathbb{P}$. We also use a selection indicator $S$ to denote whether a unit $i$ is selected into $\mathcal{D}_{\text{obs}}$, i.e., $s_i = 1$ if $\{\mathbf{x}_i, t_i, y_i\} \in \mathcal{D}_{\text{obs}}$.

Under the potential outcome framework (Imbens & Rubin, 2015), we define the potential outcomes under treatment as $Y(1)$ and under control as $Y(0)$. With the above datasets, our goal is to estimate the Conditional Average Treatment effect (CATE), which is defined as:

$$\tau(\mathbf{x}) = \mathbb{E}[Y(1) - Y(0) \mid \mathbf{X} = \mathbf{x}]. \tag{1}$$

For a unit $i$ with $t_i$ in $\mathcal{D}$, only the factual outcome $Y(t_i)$ is available. Therefore, to make CATE identifiable, we make the following commonly used assumptions (Imbens & Rubin, 2015):

**Stable Unit Treatment Value Assumption.** The distribution of the potential outcome of one unit is assumed to be independent of the treatment assignment of another unit.

**Overlap Assumption.** A unit has a nonzero probability of being treated, $0 < \mathbb{P}(T = 1 \mid \mathbf{X} = \mathbf{x}) < 1$.

**Unconfoundedness Assumption.** The treatments are independent of the potential outcomes given the pre-treatment variables, i.e., $Y(1), Y(0) \perp\!\!\!\perp T \mid \mathbf{X}$.

Based on the above assumptions, CATE can be estimated as:

$$\tau(\mathbf{x}) = \mathbb{E}[Y \mid \mathbf{X} = \mathbf{x}, T = 1] - \mathbb{E}[Y \mid \mathbf{X} = \mathbf{x}, T = 0]. \tag{2}$$

Because the sample selection mechanism is not random but is jointly determined by $T$, $\mathbf{X}$ and $Y$, $\mathbb{P}_{\{\mathbf{x},t,y\} \sim \mathcal{D}_{\mathrm{obs}}}(\mathbf{x}, t, y) \neq \mathbb{P}_{\{\mathbf{x},t,y\} \sim \mathcal{D}}(\mathbf{x}, t, y)$, i.e. $\mathbb{P}(\mathbf{X}, T, Y \mid S = 1) \neq \mathbb{P}(\mathbf{X}, T, Y)$, resulting in collider bias, which hurts CATE estimation in two aspects:

- **Biased estimation.** $Y(1), Y(0) \not\perp\!\!\!\perp T \mid \mathbf{X}, S$ because both $T$ and $Y$ cause $S$, which means the unconfoundedness assumption is no longer satisfied in $\mathcal{D}_{\mathrm{obs}}$, leading to a biased estimate of CATE using only $\mathcal{D}_{\mathrm{obs}}$.
- **Distribution shift.** $\mathbb{E}[Y \mid \mathbf{X} = \mathbf{x}, T = t, S = 1] \neq \mathbb{E}[Y \mid \mathbf{X} = \mathbf{x}, T = t]$ because $\mathbb{P}(\mathbf{X}, T, Y \mid S = 1) \neq \mathbb{P}(\mathbf{X}, T, Y)$, making the estimated CATE not only biased on the observational data $\mathcal{D}_{\mathrm{obs}}$, but also inaccurate on the true data $\mathcal{D}$.

Previous works on sample selection bias aim to either model the sample selection mechanism for reweighting or regression adjustment (Heckman, 1979; Cole & Stuart, 2010) or find variable sets that satisfy the selection-backdoor criterion for backdoor adjustment (Bareinboim & Tian, 2015). However, they cannot solve collider bias in our scenario where $T$, $\mathbf{X}$ and $Y$ all cause $S$ even $\mathcal{D}_{\mathrm{exp}}$ that satisfies $\mathbb{P}_{\mathbf{x} \sim \mathcal{D}_{\mathrm{exp}}}(\mathbf{x}) = \mathbb{P}_{\mathbf{x} \sim \mathcal{D}}(\mathbf{x})$ is available. This is because in $\mathcal{D}_{\mathrm{obs}}$, samples with $S = 0$ are missing, and for all units in $\mathcal{D}_{\mathrm{exp}}$, $S$ labels are missing. As a result, we can neither estimate $\mathbb{P}(S, Y \mid \mathbf{X} = \mathbf{x}, T = t)$ nor find variables that satisfy the backdoor criterion. Using only $\mathcal{D}_{\mathrm{exp}}$ to estimate the outcomes or generate more samples for estimation directly is not applicable either because the sample size of $\mathcal{D}_{\mathrm{exp}}$ is too small, which makes the model suffer from the severe overfitting problem. Therefore, we need to leverage both $\mathcal{D}_{\mathrm{obs}}$ and $\mathcal{D}_{\mathrm{exp}}$ to help solve collider bias.

## 3.2 Motivation

To address collider bias, we formulate it as an Out-of-Distribution (OOD) problem, as shown in Figure 1. We notice that the non-random sample selection caused by collider bias mainly results in the $S = 0$ data, i.e., the unselected data, completely missing in $\mathcal{D}_{\mathrm{obs}}$; And the critical problem in $\mathcal{D}_{\mathrm{exp}}$ is that the selection indicators are unknown. Therefore, we consider the selection indicators $S$ as the environment labels. In this way, the observational data can be regarded as samples from an environment labeled with $S = 1$, the missing unselected data can be regarded as samples from an environment labeled with $S = 0$, and the experimental data can be regarded as samples from the entire data space but the environment labels are unknown. From an OOD perspective, we wish to recover the distribution of $\mathcal{D}$ from $\mathcal{D}_{\mathrm{obs}}$ and $\mathcal{D}_{\mathrm{exp}}$ as much as possible, which means we need to recover the missing parts of $\mathcal{D}_{\mathrm{obs}}$ and $\mathcal{D}_{\mathrm{exp}}$ by two generators respectively:

- **Unselected samples generator** $\mathrm{G_d}$. It generates the missing $S = 0$ data in the observational dataset from the experimental dataset.
- **Selection indicator generator** $\mathrm{G_s}$. It generates the missing $S$ labels in the experimental dataset from the observational dataset.

To optimize the above generators, we need a discriminator to align the distribution of the generated $S = 0$ data and that of the data in $\mathcal{D}$ with $S = 0$. Because we cannot observe $S = 0$ data in $\mathcal{D}$, we use data in $\mathcal{D}_{\mathrm{exp}}$ with generated $S = 0$ labels as an approximation since $\mathcal{D}_{\mathrm{exp}}$ satisfies $\mathbb{P}_{\{\mathbf{x},t,y\} \sim \mathcal{D}_{\mathrm{exp}}}(\mathbf{x}, t, y) = \mathbb{P}_{\{\mathbf{x},t,y\} \sim \mathcal{D}}(\mathbf{x}, t, y)$. However, since the data corresponding to the two distributions we need to align is either directly or indirectly generated by the generators, the effectiveness of this discriminator depends entirely on the performance of the generators, which means it is not sufficient to achieve the objective of distribution alignment. Therefore, we introduce an additional discriminator that leverages supervised information of $\mathcal{D}_{\mathrm{obs}}$ to help align the distribution, and the two discriminators perform the following tasks respectively:

- **Selected data discriminator** $\mathrm{D_o}$. It makes the distribution of $\mathcal{D}_{\mathrm{obs}}$ the same as that of data in $\mathcal{D}_{\mathrm{exp}}$ with $S = 1$ labels generated by $\mathrm{G_s}$.

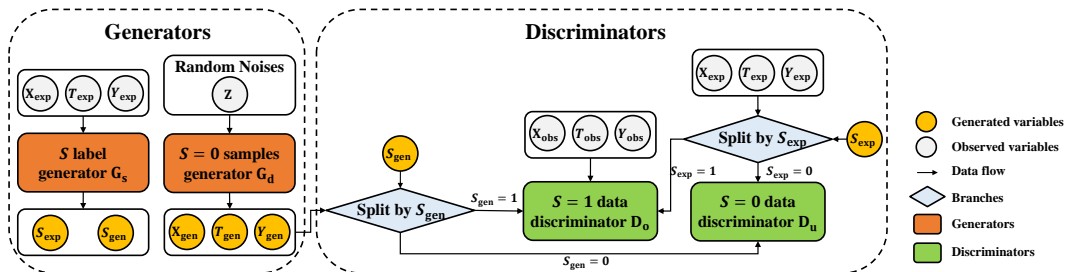

Figure 2: Overview of DCGM architecture.

- **Unselected data discriminator** $D_u$. It makes the distribution of the $S = 0$ samples generated by $G_d$ the same as that of data in $\mathcal{D}_{exp}$ with $S = 0$ labels generated by $G_s$.

To further ensure that the distribution of the combination of the generated $S = 0$ samples and $\mathcal{D}_{obs}$ is the same as that of $\mathcal{D}$, we also need an additional constraint during optimization to make the ratio of the generated samples of the observational data the same as that of the experimental data with generated $S = 0$ labels to the experimental data with generated $S = 1$ labels.

By jointly optimizing the two generators and the two discriminators with the above constraint, we can achieve the objective of recovering the distribution of $\mathcal{D}$ by combining the generated $S = 0$ samples with the original observational data, which can be used as the training data for any existing treatment effect estimation methods to achieve a better CATE estimate. Naturally, a Generative Adversarial Nets (GAN) (Goodfellow et al., 2014) based framework can achieve this optimization task.

## 3.3 DCGM: DUAL COUNTERFACTUAL GENERATIVE MODEL

Based on the above motivation, we propose a novel method named Dual Counterfactual Generative Model (DCGM), as shown in Figure 2. DCGM consists of two generators $G_d$ and $G_s$ and two discriminators $D_o$ and $D_u$, as mentioned earlier. The details are as follows:

**Unselected samples generator** $G_d$. This generator aims to generate samples whose distribution is the same as that of $S = 0$ data in $\mathcal{D}$. $G_d$ takes random noises $Z = \{z_i \sim \mathcal{N}(0, 1)\}_{i=1}^{n_{gen}}$ as inputs to generate $\mathcal{D}_{gen} = \{\mathbf{x}_i, t_i, y_i\}_{i=1}^{n_{gen}}$, denoted as $\{G_d(z_i)\}_{i=1}^{n_{gen}}$, where $n_{gen}$ is the size of the generated samples. The objective is to optimize $G_d$ to make $\mathbb{P}_{\{\mathbf{x},t,y\} \sim \mathcal{D}_{gen} \cup \mathcal{D}_{obs}}(\mathbf{x}, t, y) = \mathbb{P}_{\{\mathbf{x},t,y\} \sim \mathcal{D}}(\mathbf{x}, t, y)$, i.e., to minimize the distance between $\mathbb{P}_{z \sim \mathcal{N}(0,1)^{n_{gen}}}(G_d(z))$ and $\mathbb{P}_{\{\mathbf{x},t,y,s\} \sim \mathcal{D}}(\mathbf{x}, t, y \mid s = 0)$. Because we cannot observe $S = 0$ labeled data in $\mathcal{D}$, we cannot directly optimize $G_d$ with the above objective. Instead, we achieve this objective through two discriminators.

**Selection indicator generator** $G_s$. This generator aims to generate selection indicators $S$ for data in $\mathcal{D}_{exp}$. $G_s$ takes $(\mathbf{x}, t, y) \sim \mathcal{D}_{exp} \cup \mathcal{D}_{gen}$ as inputs to generate the corresponding $S$ labels, denoted as $G_s(\mathbf{x}, t, y)$. The objective is to optimize $G_s$ to maximize the probability of correctly labeling the data with $S$, i.e., to make $\mathbb{P}_{\{\mathbf{x},t,y\} \sim \mathcal{D}_{exp}}(\mathbf{x}, t, y, G_s(\mathbf{x}, t, y)) = \mathbb{P}_{\{\mathbf{x},t,y,s\} \sim \mathcal{D}}(\mathbf{x}, t, y \mid s)$. The key problem is that $\mathbb{P}_{\{\mathbf{x},t,y,s\} \sim \mathcal{D}}(\mathbf{x}, t, y \mid s)$ is unknown because we can only observe $S = 1$ labeled data. As a result, we cannot directly optimize $G_s$ with the above objective either but use two discriminators instead.

**Selected data discriminator** $D_o$. This discriminator aims to discriminate between data with generated $S = 1$ labels in $\mathcal{D}_{exp}$ and the observed $S = 1$ data in $\mathcal{D}_{obs}$. Following (Goodfellow et al., 2014), we regard $\mathcal{D}_{obs}$ as the original dataset, and data in $\mathcal{D}_{exp}$ and $\mathcal{D}_{gen}$ labeled with $S = 1$ by $G_s$ as the generated dataset. Therefore, $D_o$ takes $(\mathbf{x}, t, y) \sim \mathcal{D}_{obs} \cup ((\mathcal{D}_{exp} \cup \mathcal{D}_{gen}) \mid G_s(\mathbf{x}, t, y) = 1)$ as inputs and returns the probability that $(\mathbf{x}, t, y)$ is from $\mathcal{D}_{obs}$, denoted as $D_o(\mathbf{x}, t, y)$. The objective is to optimize $D_o$ to maximize the probability of correctly determining whether a sample comes from $\mathcal{D}_{obs}$ or $(\mathcal{D}_{exp} \cup \mathcal{D}_{gen}) \mid G_s(\mathcal{D}_{exp} \cup \mathcal{D}_{gen}) = 1$. The objective function is as follows:

$$\min_{G_s, G_d} \max_{D_o} \mathbb{E}_{\{\mathbf{x},t,y\} \sim \mathcal{D}_{obs}}\left[\log(D_o(\mathbf{x}, t, y))\right] + \mathbb{E}_{\{\mathbf{x},t,y\} \sim \mathcal{D}_{exp}}\left[G_s(\mathbf{x}, t, y) \cdot \log(1 - D_o(\mathbf{x}, t, y))\right]$$
$$+ \mathbb{E}_{z \sim \mathcal{N}(0,1)^{n_{gen}}}\left[G_s(G_d(z)) \cdot \log(1 - D_o(G_d(z)))\right].$$

**Unselected data discriminator** $D_u$. This discriminator aims to discriminate between the generated $S = 0$ samples and the $S = 0$ data in $\mathcal{D}$. However, since we cannot observe $S = 0$ data in $\mathcal{D}$, we can

only use data with generated $S = 0$ labels as approximations. We regard data from $\mathcal{D}_{\text{gen}}$ and $\mathcal{D}_{\text{exp}}$ labeled with $S = 0$ by $G_s$ as the original dataset, and all data from $\mathcal{D}_{\text{gen}}$ as the generated dataset. Therefore, $D_u$ takes $(\mathbf{x}, t, y) \sim \mathcal{D}_{\text{gen}} \cup (\mathcal{D}_{\text{exp}} \mid G_s(\mathbf{x}, t, y) = 0)$ as inputs and returns the probability that $(\mathbf{x}, t, y)$ is from $(\mathcal{D}_{\text{gen}} \cup \mathcal{D}_{\text{exp}}) \mid G_s(\mathbf{x}, t, y) = 0$, denoted as $D_u(\mathbf{x}, t, y)$. The objective is to optimize $D_u$ to maximize the probability of correctly determining whether a sample comes from $(\mathcal{D}_{\text{exp}} \cup \mathcal{D}_{\text{gen}}) \mid G_s(\mathcal{D}_{\text{exp}} \cup \mathcal{D}_{\text{gen}}) = 0$ or $\mathcal{D}_{\text{gen}}$. The objective function is as follows:

$$\min_{G_s, G_d} \max_{D_u} \mathbb{E}_{z \sim \mathcal{N}(0,1)^{n_{\text{gen}}}} \big[ \log(1 - D_u(G_d(z))) \big] + \mathbb{E}_{z \sim \mathcal{N}(0,1)^{n_{\text{gen}}}} \big[ (1 - G_s(G_d(z))) \cdot \log(D_u(G_d(z))) \big]$$
$$+ \mathbb{E}_{\{\mathbf{x},t,y\} \sim \mathcal{D}_{\text{exp}}} \big[ (1 - G_s(\mathbf{x}, t, y)) \cdot \log(D_u(\mathbf{x}, t, y)) \big].$$

Following (Goodfellow et al., 2014), with the above objective functions, the discriminators $D_o$, $D_u$ and the generators $G_s$, $G_d$ can be iteratively optimized using mini-batch gradient descent. In each batch, we first fix the parameters of both generators to optimize both discriminators simultaneously, then fix the parameters of both discriminators to optimize both generators simultaneously. Specifically, when fixing the parameters of the generators, the two objective functions are equivalent to minimize the following loss functions simultaneously:

$$L_{D_o} = -\mathbb{E}_{\{\mathbf{x},t,y\} \sim \mathcal{D}_{\text{obs}}} \big[ \log(D_o(\mathbf{x}, t, y)) \big] - \mathbb{E}_{\{\mathbf{x},t,y\} \sim \mathcal{D}_{\text{exp}}} \big[ G_s(\mathbf{x}, t, y) \cdot \log(1 - D_o(\mathbf{x}, t, y)) \big]$$
$$- \mathbb{E}_{z \sim \mathcal{N}(0,1)^{n_{\text{gen}}}} \big[ G_s(G_d(z)) \cdot \log(1 - D_o(G_d(z))) \big],$$
$$L_{D_u} = -\mathbb{E}_{z \sim \mathcal{N}(0,1)^{n_{\text{gen}}}} \big[ \log(1 - D_u(G_d(z))) \big] - \mathbb{E}_{z \sim \mathcal{N}(0,1)^{n_{\text{gen}}}} \big[ (1 - G_s(G_d(z))) \cdot \log(D_u(G_d(z))) \big]$$
$$- \mathbb{E}_{\{\mathbf{x},t,y\} \sim \mathcal{D}_{\text{exp}}} \big[ (1 - G_s(\mathbf{x}, t, y)) \cdot \log(D_u(\mathbf{x}, t, y)) \big],$$

and we train the discriminators by minimizing $L_{D_o} + L_{D_u}$. Given the parameters of the discriminators, the two objective functions are equivalent to minimize the following loss functions simultaneously:

$$L_{G_s} = \mathbb{E}_{z \sim \mathcal{N}(0,1)^{n_{\text{gen}}}} \big[ (1 - G_s(G_d(z))) \cdot \log(D_u(G_d(z))) \big] + \mathbb{E}_{\{\mathbf{x},t,y\} \sim \mathcal{D}_{\text{exp}}} \big[ G_s(\mathbf{x}, t, y) \cdot \log(1 - D_o(\mathbf{x}, t, y)) \big]$$
$$+ \mathbb{E}_{\{\mathbf{x},t,y\} \sim \mathcal{D}_{\text{exp}}} \big[ (1 - G_s(\mathbf{x}, t, y)) \cdot \log(D_u(\mathbf{x}, t, y)) \big] + \mathbb{E}_{z \sim \mathcal{N}(0,1)^{n_{\text{gen}}}} \big[ G_s(G_d(z)) \cdot \log(1 - D_o(G_d(z))) \big],$$
$$L_{G_d} = \mathbb{E}_{z \sim \mathcal{N}(0,1)^{n_{\text{gen}}}} \big[ G_s(G_d(z)) \cdot \log(1 - D_o(G_d(z))) \big] + \mathbb{E}_{z \sim \mathcal{N}(0,1)^{n_{\text{gen}}}} \big[ \log(1 - D_u(G_d(z))) \big]$$
$$+ \mathbb{E}_{z \sim \mathcal{N}(0,1)^{n_{\text{gen}}}} \big[ (1 - G_s(G_d(z))) \cdot \log(D_u(G_d(z))) \big],$$

and we train the generators by minimizing $L_{G_s} + L_{G_d}$. We iteratively optimize the discriminators and the generators and update $n_{\text{gen}}$ with $\frac{n_{\text{obs}} \cdot n_0}{n_1}$, where $n_{\text{obs}}$ is the sample size of $\mathcal{D}_{\text{obs}}$, $n_0$ and $n_1$ is the count of units in $\mathcal{D}_{\text{exp}}$ with $G_s(\mathbf{x}, t, y) = 0$ and $G_s(\mathbf{x}, t, y) = 1$ respectively. The iteration terminates when the maximum number of iterations reaches or the distance between $\mathbb{P}_{\{\mathbf{x},t,y\} \sim \mathcal{D}_{\text{exp}}}(\mathbf{x}, t, y)$ and $\mathbb{P}_{\{\mathbf{x},t,y\} \sim \mathcal{D}_{\text{obs}} \cup \mathcal{D}_{\text{gen}}}(\mathbf{x}, t, y)$ is less than a given threshold. Combining the generated samples $\mathcal{D}_{\text{gen}}$ and the observational data $\mathcal{D}_{\text{obs}}$, we then fit a based CATE estimator to achieve CATE estimation. Note that DCGM can be flexibly plugged into any existing CATE estimator to further estimate treatment effects. The pseudo-code of DCGM is in Appendix A.1.

## 4 EXPERIMENTS

### 4.1 BASELINES

As mentioned above, the samples generated by our proposed DCGM can be used in any existing CATE estimator to achieve better performance. To evaluate the effectiveness of the proposed method, we use Balancing Neural Network (BNN) (Johansson et al., 2016), Treatment-Agnostic Representation Network (TARNet) and CounterFactual Regression (CFR) (Shalit et al., 2017) as our based estimators. We compare the proposed method with the following baselines: (1) Doubly Robust (Bang & Robins, 2005), (2) Causal Forest (Wager & Athey, 2018), (3) Causal Effect Variational Autoencoder (CEVAE) (Louizos et al., 2017), (4) Generative Adversarial Nets for inference of Individualized Treatment Effects (GANITE) (Yoon et al., 2018), (5) BNN, (6) TARNet and (7) CFR. Based on the estimated CATE, we use the Precision in Estimation of Heterogeneous Effect (PEHE) (Shalit et al., 2017; Louizos et al., 2017) to evaluate the performance of the above methods, where PEHE $= \frac{1}{N} \sum_{i=1}^{N} \big( (\hat{y}_i(1) - \hat{y}_i(0)) - (y_i(1) - y_i(0)) \big)^2$. Note that we use the Wasserstein distance (Cuturi & Doucet, 2014) as the Integral Probability Metric (IPM) to implement BNN and CFR. We implement the baselines in the PyTorch environment with Python 3.9, with the CPU being 13th Gen Intel(R) Core(TM) i7-13700K and the GPU being NVIDIA GeForce RTX 3080 with CUDA 12.1, and we split each dataset into 60/20/20 train/validation/test datasets.

Table 1: The results (mean ± std of $\sqrt{\text{PEHE}}$) of treatment effect estimation on synthetic data.

| Estimator | $S = 1$ samples | $S = 0$ samples |
|---|---|---|
| Regression on $\mathcal{D}_{\text{exp}}$ | 4.265±0.747 | 3.651±0.170 |
| Doubly Robust | 7.410±4.602 | 8.496±2.995 |
| Causal Forest | 4.929±0.073 | 6.153±0.074 |
| CEVAE | 4.051±0.047 | 5.296±0.026 |
| GANITE | 4.139±0.071 | 4.997±0.148 |
| BNN | 2.893±0.427 | 3.196±0.360 |
| TARNet | 2.023±0.223 | 2.830±0.261 |
| CFR | 2.035±0.054 | 2.923±0.077 |
| DCGM+BNN | 0.898±0.028 | 1.123±0.036 |
| DCGM+TARNet | **0.826±0.079** | **1.022±0.145** |
| DCGM+CFR | 0.933±0.053 | 1.185±0.098 |

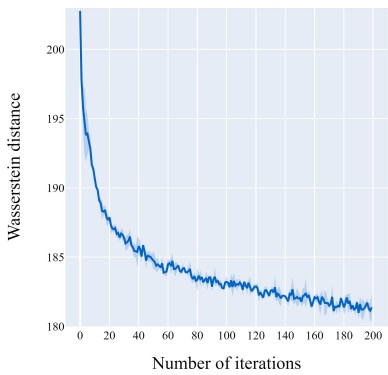

Figure 3: Wasserstein distance between data from $\mathcal{D}_{\text{exp}}$ and data from $\mathcal{D}_{\text{gen}} \cup \mathcal{D}_{\text{obs}}$.

## 4.2 EXPERIMENTS ON SYNTHETIC DATA

### 4.2.1 DATASETS

In order to evaluate the effectiveness of our method against collider bias, we generate a large-scale confounding biased dataset and a small-scale experimental dataset with the same outcome models. Specifically, we first generate continuous pre-treatment variables $\mathbf{X} \in \mathbb{R}^{n \times d}$ with independent Gaussian distributions as $\mathbf{X} \overset{\text{i.i.d.}}{\sim} \mathcal{N}(\mathbf{0}, \mathbf{1})$, where $d = 20$. For the confounding biased data, we generate binary treatments $T \in \mathbb{R}^{n \times 1}$ from a logistic function as $T \sim \text{Bernoulli}(1/(1 + e^{-t(\mathbf{X})}))$, where $\text{Bernoulli}(\cdot)$ denotes the Bernoulli distribution, $t(\mathbf{X}) = \sum_{i=1}^{d}(\mathbf{1}(\text{mod}(i, 2) \equiv 1) - \mathbf{1}(\text{mod}(i, 2) \neq 1)) \cdot (\text{mod}(i, 2) + 1) \cdot X_i/d) + \epsilon_t$, $\mathbf{1}(\cdot)$ is the indicator function, function $\text{mod}(a, b)$ returns the modulus after division of $a$ by $b$ and $\epsilon_t \sim \mathcal{N}(0, 1)$. For the experimental data, we generate treatments simply by $T \sim \text{Bernoulli}(0.5)$. Next, we generate continuous outcomes $Y \in \mathbb{R}^{n \times 1}$ from a non-linear function as $Y = T + \sum_{i=1}^{d}(T \cdot X_i + (\mathbf{1}(\text{mod}(i, 2) \neq 1) - \mathbf{1}(\text{mod}(i, 2) \equiv 1)) \cdot (\text{mod}(i, 2) + 1) \cdot (X_i + X_i^2)/d) + \epsilon_y$, where $\epsilon_y \sim \mathcal{N}(0, 1)$. To further introduce collider bias, we sample the confounding biased data by a binary selection variable $S \in \mathbb{R}^{n \times 1}$, which comes from a logistic function as $S \sim \text{Bernoulli}(1/(1 + e^{-s(\mathbf{X}, T)}))$, where $s(\mathbf{X}, T) = Y - 3 \cdot T + \sum_{i=1}^{d}(\mathbf{1}(\text{mod}(i, 2) \equiv 1) - \mathbf{1}(\text{mod}(i, 2) \neq 1)) \cdot X_i/d) + \epsilon_s$, $\epsilon_s \sim \mathcal{N}(0, 1)$ and a unit is selected into the sample only when $S = 1$. The ground truth CATE can be calculated easily by the above functions.

### 4.2.2 RESULTS

To demonstrate the effectiveness of the proposed method, we first use 500 experimental data and 10000 observational data to generate $S = 0$ samples by DCGM and then use only the observational data for training and validation of the baselines and use a combination of the observational data and generated samples for training and validation of the based estimator we choose. We also use only the experimental data for regression to estimate the CATE. We independently performed 20 experiments and regenerated the dataset for each experiment. We compare their performance and report the mean and standard deviation (std) of $\sqrt{\text{PEHE}}$ on $S = 1$ and $S = 0$ data separately, as shown in Table 1. The results show that the performance of only using the experimental data for regression is not good because of the overfitting problem. Only using observed samples with collider bias for training has poor performance on both $S = 1$ and $S = 0$ data for all baselines, among which the performance of Doubly Robust and Causal Forest is even worse than only using the experimental data for regression due to severe collider bias in observational data. Note that the performance on $S = 0$ data is inferior to that on $S = 1$ data for all estimators because of the distribution shift problem caused by collider bias. Using the samples generated by our method for training achieves significant performance improvement on all based estimators. It proves that our proposed method can effectively address collider bias and achieve more accurate CATE estimation.

To further evaluate the effectiveness of our method under different proportions of experimental data, we conduct ablations with three different sample sizes of the observational data and experimental data, namely $\{10000, 500\}$, $\{10000, 200\}$ and $\{10000, 100\}$. We conduct experiments using only

Table 2: The results (mean ± std of $\sqrt{\text{PEHE}}$) under different proportions of experimental data.

| Data+Estimator | $n_{\text{obs}} : n_e = 10000 : 500$ | | $n_{\text{obs}} : n_e = 10000 : 200$ | | $n_{\text{obs}} : n_e = 10000 : 100$ | |
|---|---|---|---|---|---|---|
| | $S = 1$ samples | $S = 0$ samples | $S = 1$ samples | $S = 0$ samples | $S = 1$ samples | $S = 0$ samples |
| Regression on $\mathcal{D}_{\text{exp}}$ | 4.265±0.747 | 3.651±0.170 | 4.393±0.470 | 3.685±0.408 | 4.652±0.602 | 3.967±0.350 |
| $\mathcal{D}_{\text{obs}}$+BNN | 2.893±0.427 | 3.196±0.360 | 2.889±0.488 | 3.167±0.201 | 2.934±0.926 | 3.448±0.664 |
| $\mathcal{D}_{\text{obs}}\&\mathcal{D}_{\text{exp}}$+BNN | 2.159±0.073 | 3.049±0.111 | 2.158±0.070 | 2.971±0.088 | 2.178±0.078 | 3.007±0.075 |
| DCGM+BNN | **0.898±0.028** | **1.123±0.036** | **1.186±0.134** | **1.559±0.157** | **1.361±0.119** | **1.857±0.103** |
| $\mathcal{D}_{\text{obs}}$+TARNet | 2.023±0.223 | 2.830±0.261 | 1.903±0.237 | 2.751±0.382 | 1.813±0.221 | 2.580±0.397 |
| $\mathcal{D}_{\text{obs}}\&\mathcal{D}_{\text{exp}}$+TARNet | 1.704±0.076 | 2.206±0.130 | 1.787±0.240 | 2.512±0.377 | 2.094±0.039 | 2.892±0.149 |
| DCGM+TARNet | **0.826±0.079** | **1.022±0.145** | **1.242±0.349** | **1.695±0.620** | **1.459±0.480** | **1.948±0.754** |
| $\mathcal{D}_{\text{obs}}$+CFR | 2.035±0.054 | 2.923±0.077 | 1.896±0.244 | 2.686±0.304 | 2.107±0.147 | 2.934±0.151 |
| $\mathcal{D}_{\text{obs}}\&\mathcal{D}_{\text{exp}}$+CFR | 1.646±0.055 | 2.214±0.097 | 1.903±0.070 | 2.634±0.052 | 2.262±0.196 | 3.043±0.137 |
| DCGM+CFR | **0.933±0.053** | **1.185±0.098** | **1.316±0.415** | **1.770±0.577** | **1.464±0.414** | **1.886±0.591** |

the experimental data to estimate CATE, using only the observational data to estimate CATE with different estimators, using a combination of the experimental data and the observational data to estimate CATE with different estimators, and using a combination of the experimental data, the observational data and the samples generated by our method to estimate CATE with different based estimators. The results are shown in Table 2: As the proportion of experimental data decreases, the performance of using the experimental data, using both the experimental data and the observational data, as well as using a combination of the experimental data, the observational data, and the generated samples gets worse for all based estimators. However, in all settings, combining the experimental data, the observational data, and the samples generated by our methods to estimate CATE achieves significant improvement in performance. Note that the variance of our method increases as the number of samples in the experimental dataset decreases because our method is based on generative models, which are known to be hard to train, especially with small-size high-dimensional data. Therefore, when the number of samples is smaller, the generative models become more challenging to train, resulting in a relatively more significant standard deviation of the final result. However, the performance of DCGM is still much better than the baselines. It proves the robustness of our method in scenarios where the experimental data is hard to obtain. We provide more ablation studies that demonstrate the necessity of each module in the framework of DCGM in Appendix A.2.

Note that the objective of DCGM is to generate $S = 0$ samples $\mathcal{D}_{\text{gen}}$ such that combining with the observational data $\mathcal{D}_{\text{obs}}$, the distribution of which is the same as that of $\mathcal{D}$. To demonstrate that DCGM indeed achieves the objective, we visualize the Wasserstein distance between $\mathbb{P}_{\{\mathbf{x},t,y\}\sim\mathcal{D}_{\text{obs}}\cup\mathcal{D}_{\text{gen}}}(\mathbf{x},t,y)$ and $\mathbb{P}_{\{\mathbf{x},t,y\}\sim\mathcal{D}_{\text{exp}}}(\mathbf{x},t,y)$ in the training process since $\mathbb{P}_{\{\mathbf{x},t,y\}\sim\mathcal{D}_{\text{exp}}}(\mathbf{x},t,y) = \mathbb{P}_{\{\mathbf{x},t,y\}\sim\mathcal{D}}(\mathbf{x},t,y)$, as shown in the Figure 3. The Wasserstein distance is an IPM measure of distance between two distributions; the smaller the Wasserstein distance, the more similar the two distributions are. It can be seen that as the number of iterations increases, the Wasserstein distance gradually decreases, proving that the optimization process of DCGM can achieve the above objective.

## 4.3 EXPERIMENTS ON REAL-WORLD DATA

### 4.3.1 DATASETS

**IHDP dataset:** The original experimental data of the Infant Health and Development Program (IHDP) aims to evaluate the effect of specialist home visits on the future cognitive test scores of premature infants (Brooksgunn et al., 1992). Following previous studies Hill (2011); Shalit et al. (2017), we remove a non-random subset of the treated group and use simulated outcomes[1] to introduce confounding bias. To obtain the experimental data, we randomly select 30 samples from the original dataset and use their noised treated and control outcomes as the factual outcomes of the treated and control groups, respectively. To introduce collider bias into the IHDP dataset, we set $S = 0$ for $T = 0$ units that both the mother boozes and the infant's score is lower than the mean value. We sample 557 units from the $S = 1$ data as the observational dataset. Intuitively, unlike the treated group which can carefully design and regularly follow up to ensure the collection of effective test results, the control group is more likely to have sample selection bias. For those mothers with boozing problems and whose children have weaker cognitive abilities, it is more likely that they will not take their children

---

[1]The dataset is available at http://www.fredjo.com/

Table 3: The results (mean ± std of $\sqrt{\text{PEHE}}$) of treatment effect estimation on real-world datasets.

| Estimator | IHDP | | Twins | |
|---|---|---|---|---|
| | $S = 1$ samples | $S = 0$ samples | $S = 1$ samples | $S = 0$ samples |
| Regression on $\mathcal{D}_{\text{exp}}$ | 3.162±0.241 | 3.146±0.225 | 0.512±0.063 | 0.510±0.071 |
| Doubly Robust | 1.391±0.288 | 1.630±0.342 | 0.485±0.052 | 0.529±0.031 |
| Causal Forest | 1.305±0.095 | 1.490±0.114 | 0.378±0.021 | 0.421±0.010 |
| CEVAE | 3.078±0.129 | 4.397±0.140 | 0.512±0.031 | 0.537±0.043 |
| GANITE | 3.063±0.158 | 3.160±0.382 | 0.329±0.022 | 0.331±0.061 |
| BNN | 1.970±0.465 | 2.086±0.441 | 0.332±0.014 | 0.384±0.054 |
| TARNet | 2.124±0.260 | 2.147±0.225 | 0.532±0.074 | 0.534±0.086 |
| CFR | 2.278±0.306 | 2.405±0.345 | 0.435±0.038 | 0.438±0.031 |
| DCGM+BNN | **0.910±0.186** | **0.897±0.199** | **0.310±0.008** | **0.308±0.014** |
| DCGM+TARNet | 1.083±0.063 | 1.064±0.101 | 0.315±0.021 | **0.308±0.015** |
| DCGM+CFR | 1.086±0.224 | 1.070±0.202 | 0.321±0.021 | 0.311±0.009 |

to participate in the cognitive test, resulting in collider bias. The final observational dataset comprises 557 units (139 treated, 418 control), and the experimental dataset comprises 60 units (30 treated, 30 control) with 26 pre-treatment variables related to the infants and their families.

**Twins dataset:** The original data of twins birth in the USA between 1989-1991 aims at evaluating the effect of low birth weight on the mortality of infants in their first year of life (Almond et al., 2005).[2] Following (Louizos et al., 2017), we select the twins whose gender is the same and weight is less than 2000kg into records. The treatment is being the heavier one in the twins, and the outcome is the one-year mortality. Because both treated (the heavier one in the twin) and control (the lighter one in the twin) outcomes are observed, we randomly select 180 samples as our experimental data. We also use the same simulation as previous works to introduce confounding bias (Louizos et al., 2017). To introduce collider bias into the dataset, we set $S = 0$ for $T = 1$ units that both the mother uses tobacco and the twin is alive. We sample 3000 units from the $S = 1$ data as our observational dataset. Intuitively, parents seldom take relatively healthy infants to the hospital, so it is more difficult to record the data of these infants, resulting in collider bias. The final observational dataset comprises 3000 units (1348 treated, 1652 control), and the experimental dataset comprises 180 units (148 treated, 152 control) with 48 pre-treatment variables related to the twins and their families.

### 4.3.2   RESULTS

We use the same training, validation, and test methods as the experiments on synthetic data. We independently perform 20 experiments and report the mean and standard deviation (std) of $\sqrt{\text{PEHE}}$ on $S = 1$ and $S = 0$ data separately, as shown in Table 3. The results show that using a combination of the observational data and samples generated by DCGM for training achieves better performance than the baselines on both $S = 0$ and $S = 1$ data. Note that DCGM achieves the best performance regardless of which based estimator is used. It proves that the proposed method can solve collider bias in real-world scenarios and achieve a more precise treatment effect estimation.

## 5   CONCLUSION

In this paper, we focus on the collider bias problem in heterogeneous treatment effect estimation, which previous works failed to address. We propose a novel Dual Counterfactual Generative Model (DCGM) that leverages small-scale unbiased experimental data and large-scale biased observation data to estimate CATE. DCGM consists of two generators and two discriminators, which can be jointly optimized in a dual way. Combining the samples generated by DCGM with the observational data, we can fit any existing CATE estimators to achieve an accurate estimate. Experiments on synthetic and real-world data demonstrate our method's effectiveness and potential application value. One main limitation is that DCGM relies on generative models to generate unselected samples in observations. Training generative models can be difficult, particularly when dealing with complex and high-dimensional data with limited samples.

---

[2]The dataset is available at https://www.nber.org/research/data/linked-birthinfant-death-cohort-data

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

# A APPENDIX

## A.1 PSEUDO-CODE

As mentioned in Section 3, we propose a novel DCGM method, which consists of two generators that respectively generate the missing $S = 0$ samples and the missing $S$ labels, as well as two discriminators that align the distribution of the combined generated $S = 0$ samples with the observational dataset to match that of the entire data space. By optimizing the generators and discriminators, DCGM can effectively generate missing data following the original distribution. Specifically, the pseudo-code of DCGM is detailed in Algorithm 1.

---

**Algorithm 1:** Dual Counterfactual Generative Model

**Data:** the observational dataset $\mathcal{D}_{\mathrm{obs}}$, the experimental dataset $\mathcal{D}_{\mathrm{exp}}$, distance threshold $\alpha$.
**Result:** generated samples $\mathcal{D}_{\mathrm{gen}}$.
$n_{\mathrm{obs}} \leftarrow$ the sample size of $\mathcal{D}_{\mathrm{obs}}$;
$n_{\mathrm{gen}} \leftarrow n_{\mathrm{obs}}$;
initialization of parameters in $\mathrm{G_s}$, $\mathrm{G_d}$, $\mathrm{D_o}$, $\mathrm{D_u}$;
**while** *the distance between* $\mathbb{P}_{\{\mathbf{x},t,y\} \sim \mathcal{D}_{\mathrm{exp}}}(\mathbf{x},t,y)$ *and* $\mathbb{P}_{\{\mathbf{x},t,y\} \sim \mathcal{D}_{\mathrm{obs}} \cup \mathcal{D}_{\mathrm{gen}}}(\mathbf{x},t,y)$ *is greater than*
$\alpha$ *and convergence of training loss of* $\mathrm{G_s}$, $\mathrm{G_d}$, $\mathrm{D_o}$, $\mathrm{D_u}$ **do**

> use mini-batch gradient descent to iteratively optimize $\mathrm{G_s}$, $\mathrm{G_d}$, $\mathrm{D_o}$, $\mathrm{D_u}$ by
>
> ❶ $\min\limits_{\mathrm{D_o},\mathrm{D_u}} - \mathbb{E}_{\{\mathbf{x},t,y\} \sim \mathcal{D}_{\mathrm{obs}}}[\log(\mathrm{D_o}(\mathbf{x},t,y))] - \mathbb{E}_{\{\mathbf{x},t,y\} \sim \mathcal{D}_{\mathrm{exp}}}[\mathrm{G_s}(\mathbf{x},t,y) \cdot \log(1 - \mathrm{D_o}(\mathbf{x},t,y))]$
>
> $\quad - \mathbb{E}_{z \sim \mathcal{N}(0,1)^{n_{\mathrm{gen}}}}[\mathrm{G_s}(\mathrm{G_d}(z)) \cdot \log(1 - \mathrm{D_o}(\mathrm{G_d}(z)))] - \mathbb{E}_{z \sim \mathcal{N}(0,1)^{n_{\mathrm{gen}}}}[\log(1 - \mathrm{D_u}(\mathrm{G_d}(z))]$
>
> $\quad - \mathbb{E}_{z \sim \mathcal{N}(0,1)^{n_{\mathrm{gen}}}}[(1 - \mathrm{G_s}(\mathrm{G_d}(z))) \cdot \log(\mathrm{D_u}(\mathrm{G_d}(z)))] - \mathbb{E}_{\{\mathbf{x},t,y\} \sim \mathcal{D}_{\mathrm{exp}}}[(1 - \mathrm{G_s}(\mathbf{x},t,y)) \cdot \log(\mathrm{D_u}(\mathbf{x},t,y))]$
>
> ❷ $\min\limits_{\mathrm{G_s},\mathrm{G_d}} \mathbb{E}_{z \sim \mathcal{N}(0,1)^{n_{\mathrm{gen}}}}[(1 - \mathrm{G_s}(\mathrm{G_d}(z))) \cdot \log(\mathrm{D_u}(\mathrm{G_d}(z)))] + \mathbb{E}_{\{\mathbf{x},t,y\} \sim \mathcal{D}_{\mathrm{exp}}}[\mathrm{G_s}(\mathbf{x},t,y) \cdot \log(1 - \mathrm{D_o}(\mathbf{x},t,y))]$
>
> $\quad + \mathbb{E}_{\{\mathbf{x},t,y\} \sim \mathcal{D}_{\mathrm{exp}}}[(1 - \mathrm{G_s}(\mathbf{x},t,y)) \cdot \log(\mathrm{D_u}(\mathbf{x},t,y))] + \mathbb{E}_{z \sim \mathcal{N}(0,1)^{n_{\mathrm{gen}}}}[\mathrm{G_s}(\mathrm{G_d}(z)) \cdot \log(1 - \mathrm{D_o}(\mathrm{G_d}(z)))]$
>
> $\quad + \mathbb{E}_{z \sim \mathcal{N}(0,1)^{n_{\mathrm{gen}}}}[\mathrm{G_s}(\mathrm{G_d}(z)) \cdot \log(1 - \mathrm{D_o}(\mathrm{G_d}(z)))] + \mathbb{E}_{z \sim \mathcal{N}(0,1)^{n_{\mathrm{gen}}}}[\log(1 - \mathrm{D_u}(\mathrm{G_d}(z)))]$
>
> $\quad + \mathbb{E}_{z \sim \mathcal{N}(0,1)^{n_{\mathrm{gen}}}}[(1 - \mathrm{G_s}(\mathrm{G_d}(z))) \cdot \log(\mathrm{D_u}(\mathrm{G_d}(z)))]$;
>
> $n_0 \leftarrow$ the count of units in $\mathcal{D}_{\mathrm{exp}}$ with $\mathrm{G_s}(\mathbf{x},t,y) = 0$;
> $n_1 \leftarrow$ the count of units in $\mathcal{D}_{\mathrm{exp}}$ with $\mathrm{G_s}(\mathbf{x},t,y) = 1$;
> $n_{\mathrm{gen}} \leftarrow \frac{n_{\mathrm{obs}} \cdot n_0}{n_1}$;
> $\mathcal{D}_{\mathrm{gen}} \leftarrow \{\mathrm{G_d}(z_i \sim \mathcal{N}(0,1))\}_{i=1}^{n_{\mathrm{gen}}}$;

**end**

---

## A.2 ABLATION STUDIES OF EACH MODULE IN DCGM

To further demonstrate the necessity of each module in the dual framework of DCGM, we compare our **DCGM** with the following ablation version of DCGM:

- **DCGM w/o** $\mathcal{D}_{\mathrm{obs}}$, uses only the experimental dataset $\mathcal{D}_{\mathrm{exp}}$ to generate samples for estimating CATE,
- **DCGM w/o** $\mathrm{G_d}$, uses only $\mathrm{G_s}$ to generate missing $S$ labels,
- **DCGM w/o** $\mathrm{D_o}$, uses only $\mathrm{D_u}$ to optimize the generators,
- **DCGM w/o** $\mathrm{D_o}$, uses only $\mathrm{D_o}$ to optimize the generators,

Note that DCGM w/o $\mathrm{G_d}$ estimate sample selection probability for estimating CATE using IPSW (Cole & Stuart, 2010), which reweights each observational sample with its inverse probability of sample selection. Then, we conduct the experiments on the same datasets with the same based estimators as those mentioned in Section 4.1 and Section 4.2.1, and compare the performance among the above different generative approaches with different based estimators. We report the mean and standard deviation (std) of $\sqrt{\mathrm{PEHE}}$ on $S = 1$ and $S = 0$ data separately. From the experimental results shown in Table 4, we can observe that each module is essential for achieving high performance in DCGM. Removing any of the modules leads to a significant decrease in performance. The observations and detailed analysis of each module are presented below.

Table 4: The results (mean ± std of $\sqrt{\text{PEHE}}$) of different generative approaches with different based estimators.

| Methods | +BNN | | +TARNet | | +CFR | |
|---|---|---|---|---|---|---|
| | $S = 1$ samples | $S = 0$ samples | $S = 1$ samples | $S = 0$ samples | $S = 1$ samples | $S = 0$ samples |
| **DCGM** | **0.898±0.028** | **1.123±0.036** | **0.826±0.079** | **1.022±0.145** | **0.933±0.053** | **1.185±0.098** |
| w/o $\mathcal{D}_{\text{obs}}$ | 1.561±0.242 | 1.823±0.409 | 2.708±0.252 | 3.411±0.218 | 2.772±0.565 | 3.519±0.296 |
| w/o $\text{G}_{\text{d}}$ | 1.752±0.587 | 2.081±0.855 | 2.186±0.531 | 1.906±0.305 | 2.167±0.994 | 1.989±0.760 |
| w/o $\text{D}_{\text{o}}$ | 2.052±0.736 | 2.083±0.787 | 2.122±0.506 | 2.386±0.503 | 2.134±0.456 | 2.258±0.345 |
| w/o $\text{D}_{\text{u}}$ | 1.344±0.392 | 1.567±0.508 | 2.241±0.429 | 2.506±0.295 | 2.312±0.495 | 2.387±0.275 |

**Experimental data is limited, and observational data can provide more information.** Compared the results of DCGM with DCGM without $\mathcal{D}_{\text{obs}}$, we can find that using only the experimental dataset to generate samples suffers from limited information and can not provide an accurate CATE estimation. Thus, we must use observational data to supplement more observations for generating unselected samples. As mentioned in Section 3.1, although $\mathcal{D}_{\text{exp}}$ is randomly sampled from $\mathcal{D}$ and thus using only $\mathcal{D}_{\text{exp}}$ seems to be able to achieve unbiased CATE estimation because the sample size of $\mathcal{D}_{\text{exp}}$ is too small, whether using it to estimate CATE directly or to generate more samples, the performance will still suffer from severe overfitting problem. Therefore, it is necessary to use not only $\mathcal{D}_{\text{exp}}$ but also $\mathcal{D}_{\text{obs}}$ to generate samples to achieve a better CATE estimate.

**Both generators are necessary, and the absence of either generator will result in the other generator not working.** DCGM without $\text{G}_{\text{d}}$, using only $\text{G}_{\text{s}}$ for IPSW, achieves better performance on only the $S = 0$ data while the performance on the $S = 1$ data gets even worse, and the variance is very high because such reweighting-based methods suffer from inaccurate estimation of the sample selection probability. It proves the necessity of using both generators. Suppose we only use $\text{G}_{\text{d}}$ to generate $S = 0$ samples. In that case, it is not feasible because there is no $S = 0$ labeled data in both $\mathcal{D}_{\text{obs}}$ and $\mathcal{D}_{\text{exp}}$ for the discriminators to use as real samples for discrimination. If we only use $\text{G}_{\text{s}}$ to generate missing $S$ labels of $\mathcal{D}_{\text{exp}}$, the first question is what we can do with these labels, or rather, what we can do with this selection indicator generator. In fact, we can use the optimized $\text{G}_{\text{s}}$ as a sample selection probability estimator to estimate $\mathbb{P}(S = 1 \mid \mathbf{X} = \mathbf{x}, T = t, Y = y)$ of each observational sample, and use reweighting based methods such as IPSW for CATE estimation. The question now becomes whether we can achieve accurate $\mathbb{P}(S = 1 \mid \mathbf{X} = \mathbf{x}, T = t, Y = y)$ estimation using only $\text{G}_{\text{s}}$. Unlike the previous case, we do have observational data with $S = 1$ labels for the selected data discriminator $\text{D}_{\text{o}}$ to use as real $S = 1$ samples for discrimination, making $\mathbb{P}_{\{\mathbf{x},t,y\} \sim \mathcal{D}_{\text{exp}}}(\mathbf{x}, t, y \mid \text{G}_{\text{s}}(\mathbf{x}, t, y) = 1) = \mathbb{P}_{\{\mathbf{x},t,y\} \sim \mathcal{D}_{\text{obs}}}(\mathbf{x}, t, y)$. However, missing $S = 0$ labeled data makes the unselected data discriminator $\text{D}_{\text{u}}$, infeasible to achieve $\mathbb{P}_{\{\mathbf{x},t,y\} \sim \mathcal{D}_{\text{exp}}}(\mathbf{x}, t, y \mid \text{G}_{\text{s}}(\mathbf{x}, t, y) = 1) = \mathbb{P}_{\{\mathbf{x},t,y,s\} \sim \mathcal{D}}(\mathbf{x}, t, y \mid s = 0)$, leading to infeasible $\mathbb{P}(S = 1 \mid \mathbf{X} = \mathbf{x}, T = t, Y = y)$ estimation.

**Both discriminators are necessary for matching the distributions of $\mathcal{D}$ and $\mathcal{D}_{\text{obs}} \cup \mathcal{D}_{\text{gen}}$.** The performance on the $S = 0$ data of using only one of the two discriminators to optimize the generators is better than simply combining the experimental and observational datasets, as shown in Table 2. However, it is still not good and stable enough compared with DCGM. It proves the necessity of using both discriminators. As mentioned in Section 3.2 and Section 3.3, the objective of DCGM is to make the distribution of the combination of the generated $S = 0$ samples and $\mathcal{D}_{\text{obs}}$ the same as that of $\mathcal{D}$. To achieve this objective, we claim that it is necessary to use both discriminators to make the distribution of $\mathcal{D}_{\text{obs}}$ the same as that of data from $\mathcal{D}_{\text{exp}}$ with generated $S = 1$ labels, as well as the distribution of the generated $S = 0$ samples the same as that of data from $\mathcal{D}_{\text{exp}}$ with generated $S = 0$ labels. If we only use $\text{D}_{\text{u}}$ to match the distribution of the generated samples and that of data from $\mathcal{D}_{\text{exp}}$ with generated $S = 0$ labels, since the generators either directly or indirectly generate the data corresponding to the two distributions we need to match, the performance of $\text{D}_{\text{u}}$ depends entirely on the performance of the generators, which means it is not sufficient to achieve the objective of distribution alignment. If we only use $\text{D}_{\text{o}}$ to match the distribution of $\mathcal{D}_{\text{obs}}$ and that of data from $\mathcal{D}_{\text{exp}}$ with generated $S = 1$ labels, even though we can use $\text{G}_{\text{s}}$ to label the samples generated by $\text{G}_{\text{d}}$ as well to make $\text{G}_{\text{d}}$ participate in the entire optimization process simply by matching the distribution of $\mathcal{D}_{\text{obs}}$ and that of data from $\mathcal{D}_{\text{exp}} \cup \mathcal{D}_{\text{gen}}$ with generated $S = 1$ labels, it is still not enough to achieve the objective since $\text{D}_{\text{o}}$ does not have any constraints on samples generated by $\text{G}_{\text{d}}$.

The above observations and analysis demonstrate that the design of DCGM is reasonable, and each module in DCGM is practical and necessary.

### A.3 DISCUSSION ON THE DIFFERENCES BETWEEN DCGM AND PREVIOUS GENERATIVE MODEL BASED CAUSAL INFERENCE METHODS

Our method is different from previous generative-model-based causal methods (Louizos et al., 2017; Yoon et al., 2018) in the following aspects:

1) **The solved problems are different.** Previous methods use generative models to solve confounding bias, while our work focuses on collider bias, which was overlooked in previous works.
2) **The targets generated by the generated models are different.** Previous methods use generative models to generate counterfactual outcomes ($Y(1-t)$) to address confounding bias, while our work aims to use generative models to generate the $S = 0$ data ($\mathbf{X}, T, Y$) that were not selected into the observational samples. To achieve this goal, we introduce an additional experimental dataset and use generative models to generate missing $S$ labels in experimental data to generate $S = 0$ samples better.
3) **The termination conditions are different.** Previous methods only use a single generative model to generate counterfactual samples, and their termination conditions are mostly the same as those of the based generative models, such as GAN. However, our proposed model consists of two dually optimized generators and two dually optimized discriminators. Therefore, in addition to meeting the primary constraints of GAN, our model also needs to ensure that the distribution of the final generated $S = 0$ samples plus the original observational samples is the same as the whole population distribution. Therefore, the termination condition also involves constraints of the distance between the above distributions.

To the best of our knowledge, we are the first to introduce GANs to solve collider bias, which is entirely different from the previous works using GANs to solve confounding bias.

