# OpenReview forum: "When Treatment Effect Estimation Meets Collider Bias: A Dual Counterfactual Generative Approach"
_ICLR.cc/2024/Conference — Submitted to ICLR 2024_

### Official Review · Reviewer_CgS4 · 2023-10-17

**Soundness:** 2 fair
**Presentation:** 3 good
**Contribution:** 2 fair
**Rating:** 5
**Confidence:** 3

**Summary:**

This paper proposes a new method to address collider bias in causal effect estimation. The main idea is to model the collider bias as a out-of-distribution estimation problem. The algorithm uses two generators for generating the observational data and the selection indicator ($S=0,1$), and correspondingly two discriminators to discern the generated data as coming from $S=1$ or $S=0$. The generators and discriminators are adversarially trained using standard methods such as GAN. Experiments are conducted on synthetic datasets and semi-synthetic datasets with artificially induced collider bias.

**Strengths:**

1. Motivation. The motivation of this work is very good. As most work (especially in AI/ML community) on causal effect estimation focus on confounder bias, it is nice to see a work that attempt to address collider bias.
2. Writing. The structure and writing of this work is well-organized and clear.

**Weaknesses:**

1. Insufficient evaluation. The selected baseline methods/datasets are not designed for collider bias evaluation. Please see questions for more details on the baselines and datasets selection.
2. Lack of identifiability guarantee. The heuristic generator/discriminator approach does not allow the derivation of identifiability results, meaning that there is no guarantee the estimated treatment effets will be unbiased even if there are abundance of samples and appropriate assumptions.
3. The proposed approach is only applicable with a binary collider variable.

**Questions:**

1. How is the CATE estimations evaluated on Twins dataset? The Twins dataset is usually used only for evaluating ATE estimations as there is no ground truth counterfactual outcomes.
2. Have the authors compared with more recently proposed CATE/ATE estimators? For example TEDVAE [1], DR-CFR [2]. As the currently compared methods are mainly not designed for collider and are a bit outdated, comparing with more recent methods can further demonstrate if the proposed approach is effective. As we are now at ICLR 2024, it seems insufficient to have most of the baselines proposed in 2016/2017.
3. Also is there any reason for selecting the datasets used in the manuscript? IHDP itself is a semi-synthetic dataset with only one data generation process, which only covers rather limited real-world scenarios. Twins is commonly used for ATE estimation instead of CATE estimation. Datasets with more comprehensive DGPs would be desirable, such as ACIC2016.

[1] Treatment effect estimation with disentangled latent factors. AAAI 2021.

[2] Learning Disentangled Representations for CounterFactual Regression. ICLR 2020.

---

> ### Author Response · Authors · 2023-11-22
> **Response to Reviewer CgS4**
>
> **Response to Weakness 1**
>
> Thank you for your comment. However, We have to emphasize our problem formulation that, to the best of our knowledge, there are no CATE estimation datasets under collider bias, so **we choose the two datasets following previous works [1,2,3] and introduce collider bias into them.** In this way, we can evaluate the performance of these estimators under collider bias. Also, as the exsiting works on collider bias mostly need additional prior knowledge that are not satisfied in these datasets, we choose baselines for slightly different tasks, i.e., CATE estimation under confounding bias and sample selection bias caused by $X$ and $T$.
>
> **Response to Weakness 2**
>
> Thank you for raising the issue. However, our motivation is to recover the target population from the collider biased observational data with the help of a representive experimental dataset without collider bias. GAN models is just a way to implement the generation process of the missing $S$ labels in $D_{exp}$ and missing $S=0$ samples in $D_{obs}$. **As long as the generative models are well trained, the $S=0$ samples can be learned well with the information of the target population contained in $D_{exp}$.**
>
> **Response to Weakness 3**
>
> Thank you for the comment. However, we have to emphasize that the definition of $S$ is that, the collected observational data is from $S=1$ and the unobserved data is $S=0$. By the definition, **we believe $S$ can only be binary since it is just an indicator of whether a unit is selected into the observed sample**. Therefore, such binary setting is reasonable common in real-world scenarios.
>
> **Response to Question 1**
>
> The Twins dataset is used in previous CATE estimation works [2,3] because **the factual outcomes of the lighter/heavier twin are regarded as the counterfactual outcomes of the heavier/lighter twin.** Therefore, **we can calculate the ground truth CATE** by using the outcome of the heavier twin minus the outcome of the lighter twin.
>
> **Response to Question 2**
>
> We appreciate your valuable advice and added more CATE estimation baselines [4,5] as you suggested, and the results are as below.
>
> | Dataset | DRCFR [4] | TEDVAE [5] | DCGM+BNN  |  DCGM+TARNet  |  DCGM+CFR  |
> |:---------| :---------: | :---------: | :---------: | :---------: | :---------: |
> Synthetic data $S=1$  | 2.141$\pm$0.203 | 4.014$\pm$0.148 | 0.898$\pm$0.028 | **0.826$\pm$0.079** | 0.933$\pm$0.053 |
> Synthetic data $S=0$ | 2.783$\pm$0.384 |  5.397$\pm$0.220 | 1.123$\pm$0.036 | **1.022$\pm$0.145** | 1.185$\pm$0.098 |
> IHDP $S=1$ | 2.267$\pm$0.339 | 4.143$\pm$0.022 | **0.910$\pm$0.186** | 1.083$\pm$0.063 | 1.086$\pm$0.224 |
> IHDP $S=0$| 2.413$\pm$0.370 | 4.154$\pm$0.037 | **0.897$\pm$0.199** | 1.064$\pm$0.101 | 1.070$\pm$0.202 |
> Twins $S=1$  | 0.388$\pm$0.028 | 0.323$\pm$0.013 | **0.310$\pm$0.008** | 0.315$\pm$0.021 | 0.321$\pm$0.021 |
> Twins $S=0$    | 0.396$\pm$0.033 | 0.337$\pm$0.011 | **0.308$\pm$0.014** | 0.308$\pm$0.015 | 0.311$\pm$0.009 |
> |||||
>
> We report the mean $\pm$ std of $\sqrt{\mathrm{PEHE}}$. The experimental results show that our method (DCGM+existing CATE estimators) outperforms recent baseline methods on CATE estimation under collider bias.
>
> **Response to Question 3**
>
> Thank you for the comment. As stated in **Response to Weakness 1** and **Response to Question 1**, we choose the IHDP and Twins datasets as we follow previous CATE estimation works [1,2,3]. We also adopt your advice and conduct additional experiments on ACIC 2016. Due to time limitation, only some of the results have been obtained, but based on the current results, our method still achieves an improvement in performance over baselines on ACIC 2016. Further results will be updated in time.
>
> We hope the above answers can solve your problems. Thank you again for your comments and suggestions. Best wishes!
>
> [1] Shalit, U., et al. (2017). Estimating individual treatment effect: generalization bounds and algorithms. INTERNATIONAL CONFERENCE ON MACHINE LEARNING, VOL 70. 70.
>
> [2] Louizos, C., et al. (2017). Causal Effect Inference with Deep Latent-Variable Models. ADVANCES IN NEURAL INFORMATION PROCESSING SYSTEMS 30 (NIPS 2017). 30.
>
> [3] Jinsung, Y., et al. (2018). GANITE: Estimation of Individualized Treatment Effects using Generative Adversarial Nets. International Conference on Learning Representations.
>
> [4] Greiner, N. H. R. (2020). Learning Disentangled Representations for CounterFactual Regression. International Conference on Learning Representations.
>
> [5] Zhang, W., et al. (2021). Treatment Effect Estimation with Disentangled Latent Factors. Thirty-Fifth {AAAI} Conference on Artificial Intelligence, {AAAI} 2021: 10923-10930.

---

### Official Review · Reviewer_vwd1 · 2023-10-29

**Soundness:** 3 good
**Presentation:** 3 good
**Contribution:** 4 excellent
**Rating:** 6
**Confidence:** 4

**Summary:**

In this paper, the authors focus on an interesting and challenging problem of causal inference, i.e., CATE estimation under collider bias, and propose a data fusion method to address it. The authors assume that a small-scale experimental dataset without collider bias is available and formulate collider bias as an OOD problem, considering the biased observational data is from a $S=1$ labeled domain and the unbiased experimental data is from the true distribution, but the $S$ labels are missing. They propose a novel method, DCGM, that utilizes generative models to generate the missing $S$ labels of the experimental dataset and the missing $S=0$ samples of the observational dataset. With the generated samples, existing CATE estimation methods can be applied even under collider bias.

**Strengths:**

1. The studied problem is interesting and important in the causality community. Collider bias is always one of the significant challenges of causal effect estimation but was ignored in most previous works.

2. The problem formulation that considers collider bias as an OOD problem is novel. The authors rethink collider bias from an OOD perspective, which is indeed a novel idea. In such a formulation, the proposed method using generative models to generate missing parts of each dataset is reasonable.

3. The ablation studies are sufficient. The authors make several ablation studies, including comparing the results using different datasets with the same CATE estimator, comparing the results with different sample sizes of the experimental dataset, and comparing the results with or without specific modules in DCGM.

4. To address collider bias, the authors assume that a small-scale experimental dataset without collider bias is available, which can be easily obtained by conducting experiments on the target population and seems a common assumption in data fusion works. The assumptions made in the paper seem reasonable to me and are impractical in real applications.

The paper is well-organized and easy to follow. The studied problem, collider bias, is interesting and challenging. The authors propose a novel formulation of collider bias as an OOD problem and propose a method to generate unbiased samples using generative models with the help of an experimental dataset without collider bias. The assumptions and methods are clearly stated and reasonable, and the experimental settings and results are detailed. One major concern is the difficulty in the GAN-based model training.

**Weaknesses:**

As the authors stated in the Conclusion, they use a GAN-based model to generate unbiased samples, which is hard to train when the sample size is small.

**Questions:**

See weakness

---

> ### Author Response · Authors · 2023-11-22
> **Response to Reviewer vwd1**
>
> Thank you for spending much time reading the paper and giving detailed comments. Responses to some of your questions follow.
>
> **Response to Weaknesses**
>
> We agree with your comments and have conducted experiments with different sizes of experimental datasets to explore the impact of the sample size on the performance of our method. You can find the results in Table 2 on Page 8.
>
> We hope the above answers can solve your problems. Thank you again for your comments and suggestions. Best wishes!

---

### Official Review · Reviewer_eRYG · 2023-10-31

**Soundness:** 1 poor
**Presentation:** 2 fair
**Contribution:** 1 poor
**Rating:** 3
**Confidence:** 5

**Summary:**

Some critical assumptions, specifically those related to the representativeness of the experimental data, are too unrealistic and effectively circumvent the real problem. Therefore I think the framework & algorithm considered here becomes irrelevant compared to what happens in reality.

**Strengths:**

the OOD formulation of the collider bias could be interesting if not considered in the literature before, in that it could make it possible to apply some of the methods developed separately in that domain to this problem in the context of causal inference.

**Weaknesses:**

Authors claim that the sample from the experimental data faithfully represent "the entire population" P_X, which is simply incorrect and misleading. In fact external validity is arguably the biggest drawback of the experimental data, as the factors that influence both selection into the experiment (S=1) and modify the treatment effect lead to a difference in the average causal effects between the experimental population and a target populations.

In healthcare, for instance, people who are eligible AND join the trials (i.e. the experimental data in your case) are nothing near representative of the entire population. Therefore, while it is common practice to compare experimental and observational data or attempt to combine them, this is only possible, without further assumptions, to the extent of "overlap" between the entire population (as captured by the observational data) and the experimental population.  See some references for the generalizability of experiments problem in various domains:

"External validity of randomised controlled trials:“to whom do the results of this trial apply?” Rothwell et al., The Lancet, 2005.
"The generalizability of survey experiments," Mullinick et al., Journal of Experimental Political Science, 2015
"Generalizing causal inferences from individuals in randomized trials to all trial‐eligible individuals," Dahabreh et al., Biometrics. , 2019

Furthermore, in Section 3.2 the authors state:

"We notice that the non-random sample selection caused by collider bias mainly results in the S = 0 data, i.e., the unselected data, completely missing in $D_{\mathrm{obs}}$; And the critical problem in $D_{\mathrm{exp}}$ is that the selection indicators are unknown."

It should be cited/noted where did they "notice" that? I do not see how, in real-world, the experimental data is never observed in the observational data. That would indicate zero overlap between studies which essentially makes any sort of joint inference infeasible. Also why the selection indicators are unknown in $D_{\mathrm{exp}}$? Isnt it by definition $S=1$? And if experimental data is never selected as mentioned before, why would we expect the generator to ever generate S=0? I think what makes the algorithm work in experiments is the limited implicit overlap between the experimental and observational datasets that is also implied in the design of the generators, which does not seem to be inline with the verbal arguments/assumptions.

The assumptions as t how the covariate distributions in the experimental and observational distributions differ, with explicit reference to the role of selection/collider bias, should be made more clear, while staying grounded (i.e., the experimental data does not represent the entire population.)

**Questions:**

no questions other than those mentioned in Weaknesses.

---

> ### Author Response · Authors · 2023-11-22
> **Response to Reviewer eRYG**
>
> Thank you for spending much time reading the paper and giving detailed comments. Responses to some of your questions follow.
>
> **Response to Weaknesses**
>
> We agree that in some scenarios, experimental data may not be representive for the target population, and the proposed method is not applicable. However, **we think there are also many scenarios that the experimental population is the same as the target population, and such experimental design is feasible.** For example, when studying the effect of pricing on restaurant evaluation on the online ordering platform, only those who extremely like the restaurant's dishes or have a lot of complaints will take the time to evaluate the restaurant on the platform, which is the observation data. However, the target population is actually all users who use the platform, which leads to selection bias. The platform can randomly select a small number of users from among all users, and encourage them to evaluate the restaurant by discounts or coupons, which is the experimental data that can represent the target population.
>
> We have to emphasize our problem formulation that throughout the paper, **we use $S=1$ to denote whether a unit is selected into the observational data**, which is the observational population. The experimental data is from another population that we assume is the same as the target population. Therefore, **units in the experimental data are not all from the $S=1$ distribution** (the observational population).
>
> We hope the above answers can solve your problems. Thank you again for your comments and suggestions. Best wishes!

---

### Official Review · Reviewer_NFEC · 2023-11-01

**Soundness:** 3 good
**Presentation:** 3 good
**Contribution:** 3 good
**Rating:** 5
**Confidence:** 3

**Summary:**

The paper introduces DCGM, a new approach to tackle collider bias by treating the selection indicator (S) as an environment label. It utilizes both observational and experimental data to generate missing S = 0 samples for the observational dataset, generate missing S labels for the experimental dataset, and align the distribution of the generated S = 0 samples with the observational dataset. DCGM comprises two generators and two discriminators. The generators generate missing data and labels, while the discriminators distinguish between observational and generated data. By optimizing the generators with the discriminators, DCGM effectively generates missing data while preserving the original data distribution.

**Strengths:**

- The paper provides a thorough examination of various biases in treatment effect estimation, including confounding bias, selection bias, and the often overlooked collider bias.

- The paper introduces an innovative formulation of collider bias as an out-of-distribution problem, providing a fresh perspective on the issue and offering a novel approach to tackle it.

- The proposed DCGM is a novel and well-defined solution to address collider bias by generating missing data and labels. The method comprises generators and discriminators, and its effectiveness is demonstrated through extensive experiments.

**Weaknesses:**

It seems to me that D_exp contains confounding bias (Figure 1a). However, the simulation of T in synthetic data is T ∼ Bernoulli(0.5) for D_exp -- without confounding bias; meanwhile, the simulation of T in D_obs is T ∼ Bernoulli(1/(1 + e−t(X))) -- with confounding bias.

Hence, a follow-up concern is whether the proposed method is applicable in this synthetic data since the proposed method makes the distribution of D_obs the same as that of data in D_exp with S = 1. The concern is that they cannot be the same since they are sampled from two different causal graphs, i.e., there is an edge X->T for D_obs while this edge doesn't exist in D_exp.

I believe that both D_obs and D_exp should be generated with the same causal graph. In D_obs, only samples with S=1 are observed. In D_exp, all the records for both S=1 and S=0 are observed except that the label S for all records are unknown. This is why D_obs and D_exp can be aligned.

**Questions:**

Can the authors explain why there are such inconsistencies in their proposed method and the experiments (See weaknesses)?

---

> ### Author Response · Authors · 2023-11-22
> **Response to Reviewer NFEC**
>
> Thank you for spending much time reading the paper and giving detailed comments. Responses to some of your questions follow.
>
> **Response to Weaknesses**
>
> We thank the author for pointing out the issue. We agree that "the experimental data and the observational data should be generated with the same causal graph. In $D_{obs}$, only samples with $S=1$ are observed. In $D_{exp}$, all the records for both $S=1$ and $S=0$ are observed except that the label $S$ for all records are unknown." We conduct the experiments (on synthetic datasets) without confounding bias just because **we want to focus on the evaluation of CATE estimation under collider bias**, leading to possible confusion since it does not exactly matches the causal graph in Figure 1. We adopted your advice and re-ran all the experiments with a new setting of $D_{exp}$ and $D_{obs}$ that both are confounding biased. Due to time limitation, only some of the results have been obtained, but based on the current results, our method still achieves an improvement in performance over baselines in this new setting. Further results will be updated in time.
>
> **Response to Questions**
>
> Thank you for your comment. As we wrote in **Response to Weaknesses**, we adopted your advice and re-ran all the experiments with a new setting of $D_{exp}$ and $D_{obs}$ that both are confounding biased. Due to time limitation, only some of the results have been obtained, but based on the current results, our method still achieves an improvement in performance over baselines in this new setting. We sincerely appreciate your kind advice.
>
> We hope the above answers can solve your problems. Thank you again for your comments and suggestions. Best wishes!

---

> > ### Comment · Reviewer_NFEC · 2023-11-23
> > **Response to authors**
> >
> > Thank you for your response. Incorporating these changes will enhance the quality of the paper.

---

### Meta-Review · Program_Chairs · 2023-12-04

**Metareview:**

In this paper, the authors address collider bias in the observational data by introducing small-scale experimental data and propose a novel method named Dual Counterfactual Generative Model (DCGM).

A reasonable amount of discussions took place between the authors and the reviewers. In the end, we got four reviews with ratings of 5, 3, 6, and 5 with confidence of 3, 5, 4, and 3 respectively. The reviewers appreciate the novel framework and the comprehensive experiments.

However, the issues raised by the reviews are critical, including the problem/experiment setting (NFEC, eRYG), experiment result (vwd1, CgS4), assumptions (eRYG), and theoretical plausibility (eRYG, CgS4). The decision is reject.

**Justification For Why Not Higher Score:**

The issues raised by the reviews are critical and all reviewers consensus to reject.

**Justification For Why Not Lower Score:**

N/A

---

### Decision · Program_Chairs · 2024-01-16

Reject